# Degradation of Agro-Industrial Wastewater Model Compound by UV-A-Fenton Process: Batch vs. Continuous Mode

**DOI:** 10.3390/ijerph20021276

**Published:** 2023-01-10

**Authors:** Nuno Jorge, Ana R. Teixeira, José R. Fernandes, Ivo Oliveira, Marco S. Lucas, José A. Peres

**Affiliations:** 1Escuela Internacional de Doctorado (EIDO), Campus da Auga, Campus Universitário de Ourense, Universidade de Vigo, As Lagoas, 32004 Ourense, Spain; 2Centro de Química de Vila Real (CQVR), Departamento de Química, Universidade de Trás-os-Montes e Alto Douro (UTAD), Quinta de Prados, 5000-801 Vila Real, Portugal; 3Centro de Química de Vila Real (CQVR), Departamento de Física, Universidade de Trás-os-Montes e Alto Douro (UTAD), Quinta de Prados, 5000-801 Vila Real, Portugal; 4Centre for the Research and Technology of Agro-Environmental and Biological Sciences (CITAB), University of Trás-os-Montes and Alto Douro (UTAD), 5000-801 Vila Real, Portugal

**Keywords:** caffeic acid, electric energy per order, environmental impact, photo-Fenton, UV-A LEDs, winery wastewater

## Abstract

The degradation of a model agro-industrial wastewater phenolic compound (caffeic acid, CA) by a UV-A-Fenton system was investigated in this work. Experiments were carried out in order to compare batch and continuous mode. Initially, batch experiments showed that UV-A-Fenton at pH 3.0 (pH of CA solution) achieved a higher generation of ${\mathrm{HO}}^{•}$, leading to high CA degradation (>99.5%). The influence of different operational conditions, such as H_2_O_2_ and Fe^2+^ concentrations, were evaluated. The results fit a pseudo first-order (PFO) kinetic model, and a high kinetic rate of CA removal was observed, with a [CA] = 5.5 × 10^−4^ mol/L, [H_2_O_2_] = 2.2 × 10^−3^ mol/L and [Fe^2+^] = 1.1 × 10^−4^ mol/L (*k*_CA_ = 0.694 min^−1^), with an electric energy per order ($E_{\mathrm{EO}}$) of 7.23 kWh m^−3^ order^−1^. Under the same operational conditions, experiments in continuous mode were performed under different flow rates. The results showed that CA achieved a steady state with higher space-times (θ = 0.04) in comparison to dissolved organic carbon (DOC) removal (θ = 0–0.020). The results showed that by increasing the flow rate (*F*) from 1 to 4 mL min^−1^, the CA and DOC removal rate increased significantly (*k*_CA_ = 0.468 min^−1^; *k*_DOC_ = 0.00896 min^−1^). It is concluded that continuous modes are advantageous systems that can be adapted to wastewater treatment plants for the treatment of real agro-industrial wastewaters.

## 1. Introduction

The rapid expansion of the agro-industry in both developed and developing countries is a major contributor of environmental pollution worldwide [1,2]. Agro-industrial wastewater characteristics are much more diverse than domestic wastewater, which is usually qualitatively and quantitatively similar in its composition. This industry produces large quantities of highly polluted wastewater containing toxic substances and organic and inorganic compounds such as: heavy metals, pesticides, phenols, and derivatives [3,4]. In the case of industries such as wine or olive oil production, large volumes of wastewaters are produced from different activities, namely tank washing, transfer, bottling, and filtration [5,6,7]. These wastewaters are characterized by an elevated content of suspended solids, low pH (3–5), and high organic load composed of phenolic compounds, sugars, organic acids, and esters [8,9]. Among these compounds, polyphenols are considered to be hazardous compounds because they are not mineralized by conventional biologic treatments [10,11], among which polyphenolic compounds are found to be the most abundant, including gallic, syringic, protocatechuic, vanillic, and caffeic acids [12,13,14,15]. Caffeic acid (3,4-dihydroxycinnamic acid) is considered to be one of the most refractory phenolic compounds to biologic degradation, because it exhibits high toxicity and antibacterial activity and represents a risk to human health, reaching 100 mg L^−1^, with a half-life ranging from 21.2 to 26.7 min in natural conditions [12,16,17].

Due to the limitations of conventional wastewater treatment technologies in removing recalcitrant pollutants such as caffeic acid, a more effective treatment is required to achieve a complete mineralization of these organic compounds, such as advanced oxidation processes (AOPs). In the AOPs, there is the generation of hydroxyl radicals (${\mathrm{HO}}^{•}$), which react with organic compounds, oxidizing them to simpler intermediates and possibly to CO_2_ and H_2_O [18,19]. The ${\mathrm{HO}}^{•}$ radicals are advantageous because they (1) do not generate additional waste, (2) are not toxic and have a very short lifetime, (3) are not corrosive to pieces of equipment, (4) are usually produced by assemblies that are simple to manipulate, and (5) have a high oxidizing potential (E° = 2.80 V) regarding sulfate radical anion (E° = 2.60 V), ozone (E° = 2.08 V), hydrogen peroxide (E° = 1.76), and chlorine (E° = 1.36) [20,21]. Among the AOPs, the photo-Fenton process appears as a suitable treatment process for ${\mathrm{HO}}^{•}$ radical production [22]. It employs Fe^2+^ and H_2_O_2_, which are readily available, easy to handle, and environmentally benign [18] and “near-UV to visible region” of light, up to a wavelength of 600 nm [23] to improve the ${\mathrm{HO}}^{•}$ radical production and to rapidly reduce the Fe^3+^ back to Fe^2+^ [24,25]. Traditional mercury-based UV-C radiation lamps can become very unstable, due to their overheating, decreasing their efficiency and lifetime [26,27]. The UV-A LED lights are a good alternative to UV-C mercury lamps because they have longer lifetimes, lower energy consumption, higher efficiency, do not overheat, and are less harmful to the environment [28,29].

Although batch reactors have been shown to be efficient in contaminant degradation, they are expensive when upscaling the treatment of wastewater; however, to counteract this tendency, most industrial treatment facilities operate in continuous mode [30]. Therefore, the aim of this work was to evaluate the degradation of caffeic acid in batch and continuous mode, in a UV-A LED reactor with a wavelength of 365 nm. It is also intended to evaluate how the different variables (pH, concentration of H_2_O_2_, and Fe^2+^) affect the kinetic rate of CA degradation and energy consumption. The novelty of this work lies in the application of a continuous system coupled with a UV-A reactor to degrade the CA phenolic, which has never been performed before.

## 2. Materials and Methods

### 2.1. Reagents

Caffeic acid (3,4-Dihydroxycinnamic acid) was acquired from Sigma-Aldrich, St. Louis, MO, and used as received without further purification. The molecular structure of caffeic acid in non-hydrolyzed form is illustrated in Table 1. Iron (II) sulfate heptahydrate (FeSO_4_•7H_2_O) was acquired from Panreac, Barcelona, Spain, and hydrogen peroxide (H_2_O_2_ 30% *w*/*w*) and titanium (IV) oxysulfate solution 1.9–2.1%, for determination of hydrogen peroxide, were acquired from Sigma-Aldrich, St. Louis, MO, USA. Sodium sulfite anhydrous (Na_2_SO_3_) was acquired from Merk, Darmstadt, Germany. Trifluoroacetic acid (HPLC grade, ≥99.0%) was acquired from Riedel-de Haën, Seelze, Germany, and acetonitrile (HPLC grade, ≥99.9%) was acquired from Chem-Lab, Zedelgem, Belgium. For pH adjustment, sodium hydroxide (NaOH) from Labkem, Barcelona, Spain, it was used, along with sulfuric acid (H_2_SO_4_, 95%) from Scharlau, Barcelona, Spain. Deionized water was used to prepare the respective solutions.

### 2.2. Analytical Determinations

Different parameters were measured in order to determine the effect of the treatments. The dissolved organic carbon (DOC) in mg C/L and total nitrogen (TN) in mg N/L were determined by direct injection of filtered samples into a Shimadzu TOC-LCSH analyzer (Shimadzu, Kyoto, Japan), equipped with an ASI-L autosampler, provided with an NDIR detector and calibrated with standard solutions of potassium phthalate. The hydrogen peroxide concentration was determined using titanium (IV) oxysulfate (DIN 38 402H15 method) at 410 nm using a portable spectrophotometer from Hach (Loveland, CO, USA), the pH and oxidation reduction potential (ORP) were measured by a 3510 pH meter (Jenway, Cole-Parmer, UK), and the iron concentrations were analyzed by atomic absorption spectroscopy (AAS) using a Thermo Scientific™ iCE™ 3000 Series (Thermo Fisher Scientific, Waltham, MA, USA).

The caffeic acid (CA) eluting peaks were monitored at 280 nm using software Chromeleon™ 7.2.9 (Thermo Fisher Scientific, Waltham, MA, USA). The CA concentration was monitored by a UHPLC Ultimate 3000 (Thermo Fisher Scientific, Waltham, MA, USA), using a C18 reverse phase column (250 × 4.6 mm, 5 μm) with a flowrate of 1 mL min^−1^ at 25.0 °C. The volume of injection was 10.00 µL, and the eluents used were ultrapure water/trifluoroacetic acid (99.9:0.1, *v/v*) (solvent A) and acetonitrile/trifluoroacetic acid (99.9:0.1, *v/v*) (solvent B) upon the linear gradient scheme (t in min; %B): 0, 0%B; 5, 20%B; 10, 100%B; 16, 0%B, 20, 0%B.

### 2.3. Fenton-Based Experimental Procedure

The Fenton-based batch experiments were performed in a self-designed lab-scale reactor with 500 mL capacity and a solution depth of 1.4 cm (Figure 1). The lab reactor had a rectangular shape with a bottom and mirror walls. The UV-A LEDs system was composed of 12 Indium Gallium Nitride (LnGaN) LEDs lamps (Roithner AP2C1-365E LEDs) with a λ_max_ = 365 nm. 250 mL of CA solution with a concentration of 5.5 × 10^−4^ mol L^−1^ (pH = 3.90 ± 0.19, DOC = 68.7 ± 2.1 mg C L^−1^), which was constantly agitated (350 rpm) by a L32 Basic Hotplate Magnetic Stirre 20 L (Labinco, Breda, Netherlands) at ambient temperature (298 K) for 15 min. Initially, different AOPs were tested (H_2_O_2_, UV-A, Fe^2+^ + UV-A, H_2_O_2_ + UV-A, Fenton, and UV-A-Fenton), then the pH (3.0–7.0), H_2_O_2_ concentration (5.5 × 10^−4^–8.8 × 10^−3^ mol L^−1^), and Fe^2+^ concentration (0.18 × 10^−4^–11 × 10^−4^ mol L^−1^) were varied.

For the Fenton-based continuous experiments, a reservoir filled with CA with a concentration of 5.5 × 10^−4^ mol L^−1^ and H_2_O_2_ was pumped by a peristaltic pump (Shenchen, Hvidovre, Denmark) into the UV-A reactor, in which different flow rates (1–4 mL min^−1^) were applied to the aqueous mean, with a hydraulic retention time (HRT) of 15 min (Figure 1). The H_2_O_2_ was added in continuous mode to prevent radical scavenging and to enhance CA degradation.

To determine the CA and DOC removal, Equation (1) was applied [32].
(1)$$\mathrm{Removal}\;(\%)=\;\frac{C_{0}{\;-\;C}_{t}}{C_{0}}\;\times \;100$$
where C_0_ and C_t_ are the initial and final concentrations of CA and DOC.

### 2.4. Electrical Energy Determination

Assuming the degradation of CA as a pseudo-first order (PFO) kinetic model (ln[CA]_t_ = −kt + ln[CA]_0_), the electrical energy per order ($E_{\mathrm{EO}}$) can be determined by conversion of the units of the first order kinetic constants to min^−1^, which results in Equation (2) [33]:(2)$$E_{\mathrm{EO}}=\frac{38{.4\;\times \;10}^{-3}*P*1000}{{V*k}_{\mathrm{obs}}}$$
where *P* is the nominal power of the reactor (kW), V is the volume (m^3^), and k_obs_ is the pseudo-first order kinetic observed (min^−1^).

All the CA removal experiments were performed in triplicate, and the observed standard deviation was always less than 5% of the reported values. Differences among means were determined by analysis of variance (ANOVA) using OriginLab 2019 software (Northampton, MA, USA), and the Tukey’s test was used for the comparison of means, which were considerate different when *p* < 0.05, and the data are presented as mean and standard deviation (mean ± SD).

## 3. Results

### 3.1. Chemical Degradability of Caffeic Acid

Considering the high content of impurities present in agro-industrial wastewaters, it is difficult to understand the degradation of the single compounds present in their constitution. Therefore, in this work caffeic acid (CA) was selected as a model compound because it exists in many types of agro-industrial wastewater. To evaluate the capacity of the UV-A reactor for the degradation of CA phenolic, several experiments were carried out: (1) CA+H_2_O_2_, (2) CA+UV-A, (3) CA+Fe^2+^+UV-A, (4) CA+H_2_O_2_+UV-A, (5) CA+H_2_O_2_+Fe^2+^, (6) CA+H_2_O_2_+Fe^2+^+UV-A. In Figure 2a are represented the results of CA removal after each treatment. From the results, it was possible to observe that CA is resistant to oxidation with the application of H_2_O_2_, UV-A, Fe^2+^+UV-A, and H_2_O_2_+UV-A, with 1.5, 4.7, 7.9, and 12.7%, respectively. With the application of H_2_O_2_, UV-A, and Fe^2+^+UV-A, there is no generation of hydroxyl radicals (${\mathrm{HO}}^{•}$) and CA shows to be resistant to degradation by radiation, H_2_O_2_, and iron. In accordance with these results, CA is harder to degrade regarding other phenolics, such as gallic acid [34]. The combination of H_2_O_2_ + UV-A increased the degradation of CA, due to the conversion of H_2_O_2_ into ${\mathrm{HO}}^{•}$ radicals (Equation (3)). The highest CA removals were observed with application of the Fenton and UV-A-Fenton processes, with 92.3 and 99.9% after 15 min of reaction. An analysis of the ORP values showed higher values with the application of Fenton and photo-Fenton, which could be linked to the production of ${\mathrm{HO}}^{•}$ radicals. To understand these results, the conversion of the H_2_O_2_ and the concentration of Fe^2+^ available in solution were studied (Figure 2b). With the application of H_2_O_2_ and H_2_O_2_ + UV-A, there was a low consumption of H_2_O_2_ (0.24 and 0.45 mM), thus a low generation of ${\mathrm{HO}}^{•}$ radicals occurred. With application of Fenton and UV-A-Fenton, the H_2_O_2_ consumption increased (0.80 and 1.20 mM). This increase could be due to the reaction of Fe^2+^ with H_2_O_2_ (Equation (4)). This difference in H_2_O_2_ consumption could be due to the Fe^2+^ available. Due to the reaction of Fe^2+^ with H_2_O_2_, the Fe^2+^ was oxidized to Fe^3+^, precipitating and ferric hydroxide. The UV-A was able to regenerate the Fe^3+^ to Fe^2+^ (Equation (5)) [35,36], thus a higher ${\mathrm{HO}}^{•}$ radicals generation occurred.
(3)$$H_{2}O_{2}+\mathrm{UV}\;\to \;{2\mathrm{HO}}^{•}$$
(4)$$\mathrm{Fe}2^{+}+H_{2}O_{2}\;\to \;\mathrm{Fe}3^{+}+{\mathrm{HO}}^{•}+{\mathrm{HO}}^{-}$$
(5)$${\mathrm{Fe}(\mathrm{HO})}^{2+}+\mathrm{UV}\;\to \;\mathrm{Fe}2^{+}+{\mathrm{HO}}^{•}$$

These results were in agreement with the work of Cruz et al. [37], who observed an efficient removal of Sulfamethoxazole by the combination of catalyst + H_2_O_2_ + UV radiation.

### 3.2. Effect of pH

The pH of the solution has an important effect in the degradation efficiency of CA. Previous authors observed that the downward trend of ${\mathrm{HO}}^{•}$ radical production increased with higher pH, because the Fenton reaction’s optimal pH is around pH 2–4 [38]. The pH of the CA solution was varied (3.0–7.0), with results showing a CA removal of 99.9, 87.7, 43.0, and 41.3%, respectively, for pH 3.0, 4.0, 6.0, and 7.0 (Figure 3a). As the pH increased above pH > 3.0, the availability of the iron decreased (0.090, 0.079, 0.068, and 0.063 mM, respectively) (Figure 3b). At pH above 3.0, the Fe^3+^ produced by the oxidation of Fe^2+^ in the Fenton process began to precipitate in the form of amorphous Fe(HO)_3_. This formation not only decreased the iron concentration, but also inhibited the regeneration of Fe^2+^ [39]. In accordance with Equations (6) and (7), the reaction between Fe^3+^ and H_2_O_2_ generates ferric-hydroperoxyl complexes that decomposes to produce ${\mathrm{HO}}_{2}^{•}$ radicals and Fe^2+^, as shown in Equation (8); however, these reactions have a very slow reaction rate (2.7 × 10^−3^ M s^−1^) [40]. Thus, at pH > 3.0, a lower concentration of H_2_O_2_ was converted to ${\mathrm{HO}}^{•}$ radicals, which decreased the efficiency of the UV-A-Fenton process.
(6)$$\mathrm{Fe}3^{+}+H_{2}O_{2}\;\to \;{\mathrm{Fe}(\mathrm{HO}}_{2}{)}^{2+}+H^{+}$$
(7)$${\mathrm{FeHO}}^{2+}+H_{2}O_{2}\;\to \;{\mathrm{Fe}(\mathrm{HO})(\mathrm{HO}}_{2}{)}^{+}+H^{+}$$
(8)$${\mathrm{Fe}(\mathrm{HO})(\mathrm{HO}}_{2}{)}^{+}\;\to \;\mathrm{Fe}2^{+}+{\mathrm{HO}}^{-}+{\mathrm{HO}}_{2}^{•}$$

A similar result was observed in the work of Zha et al. [41], in the degradation of landfill leachate by the photo-Fenton process. Considering that the pH of winery wastewaters is usually between 3 and 4, the application of photo-Fenton at pH 3.0 decreases the requirement of reagent consumption for pH adjustments.

### 3.3. Effect of H_2_O_2_ Concentration

In this section, the effect of the oxidant (H_2_O_2_) concentration in the efficiency of the UV-A-Fenton to degrade the CA phenolic was evaluated. To keep the UV-A-Fenton competitive with other processes, it is essential that this process represents a low-cost operation, thus the control of the H_2_O_2_ concentration is implied. In observation of Figure 4a, different H_2_O_2_:CA molar ratios (R) were tested, with results showing a tendency to increase the CA removal by increasing the molar rate. With application of an R = 1 and 2, a low CA removal was observed; however, when the R = 4 was applied, a near complete removal of CA was observed, reaching a plateau after 10 min of reaction. Above this ratio, no considerable CA removal was observed. To understand these results, the H_2_O_2_ consumption and Fe^2+^ concentration, along with the reaction, were studied (Figure 4b). With application of R = 1 and 2, the H_2_O_2_ consumption was low (0.10 and 0.30 mM, respectively), despite the high concentration of Fe^2+^ available (0.088 and 0.095 mM, respectively), which could mean that the H_2_O_2_ consumed was insufficient to produce ${\mathrm{HO}}^{•}$ radicals in a necessary amount to degrade the CA phenolic. With application of R = 4, 8, and 16, the iron concentrations remained similar (0.090, 0.087, and 0.086 mM, respectively); however, a higher H_2_O_2_ consumption was observed (1.2, 1.5, and 1.9 mM), thus a higher amount of ${\mathrm{HO}}^{•}$ radicals were generated, leading to the degradation of the CA phenolic. The consumption of H_2_O_2_ within R = 8 and 16 could be linked to the scavenging reactions between the excess of H_2_O_2_ with ${\mathrm{HO}}^{•}$ (Equations (9) and (10)), leading to the production of hydroperoxyl radicals (${\mathrm{HO}}_{2}^{•}$) and superoxide anion radicals ($O_{2}^{•-}$) with a lower oxidation potential (E° = 1.70 and 0.40 V) than ${\mathrm{HO}}^{•}$ radicals (E° = 2.80 V) [42].
(9)$$H_{2}O_{2}+{\mathrm{HO}}^{•}\;\to \;H_{2}O+{\mathrm{HO}}_{2}^{•}$$
(10)$$H_{2}O_{2}+{\mathrm{HO}}^{•}\;\to \;H_{2}O+H^{+}+O_{2}^{•-}$$

To have a better understanding of the effect of the H_2_O_2_ in phenolic degradation, the results showed a good fitting to the pseudo-first order (PFO) kinetic model. The *k* values vs. [H_2_O_2_] were plotted and two different kinetic rates were separated (Figure 5a). In a first one, up to 2.2 × 10^−3^ mol L^−1^ H_2_O_2_, the PFO constant is directly proportional to the concentration of H_2_O_2_ applied, with a slope of 0.012 mol L^−1^ min^−1^. In the second kinetic rate (up to 8.8 × 10^−3^ mol L^−1^), the PFO constant increased linearly with the concentration of H_2_O_2_, with a slope of 0.489 mol L^−1^ min^−1^. Based in these results, it can be assumed that the LED light fully penetrates the solution, independent of the concentration of H_2_O_2_ used, so the geometry of the UV-A reactor and the depth of the solution appeared to be adequate. These results were shown to be in agreement with the work of Li and Cheng [43], who observed a direct relation between the increase in the kinetic rate of malachite green degradation with higher H_2_O_2_ concentrations.

It is also necessary to study the how the concentration of H_2_O_2_ affects the energy consumption of the reactor, because the energy consumption determines part of the treatment costs. In Figure 5b, the $E_{\mathrm{EO}}$ values vs. the [H_2_O_2_] were plotted, with results showing two distinct areas: (1) one for [H_2_O_2_] ≤ 2.2 × 10^−3^ mol L^−1^ and (2) one for [H_2_O_2_] ≥ 2.2 × 10^−3^ mol L^−1^. For concentrations below 2.2 × 10^−3^ mol L^−1^, the degradation of CA was too slow, which implied a higher energy consumption. For concentrations higher than 2.2 × 10^−3^ mol L^−1^, only small gains were observed, which are not profitable, considering the high cost of H_2_O_2_ [44].

### 3.4. Effect of Fe^2+^ Concentration

As previously observed in Figure 2, the Fe^2+^ catalyst had a major effect regarding UV-A radiation in the conversion of H_2_O_2_ into ${\mathrm{HO}}^{•}$ radicals. Therefore, the H_2_O_2_:Fe^2+^ molar ratio (R1) was varied to study the effect of the Fe^2+^ concentration in the UV-A-Fenton process (Figure 6a). The results showed that by increasing the catalyst concentration, the rate of the CA phenolic degradation was increased. Clearly, the application of an R1 from 120 to 40 was revealed to be insufficient to achieve a complete degradation of CA. Qualitatively, the results indicated that increasing the Fe^2+^ concentration in solution, for a constant H_2_O_2_ concentration, increased the degradation rate of CA. To understand the effect of the Fe^2+^ concentration, the H_2_O_2_ consumption and the concentration of iron that remained in solution were analyzed (Figure 6b). With the application of an R1 from 120 to 40, lower concentrations of H_2_O_2_ were consumed, regarding the values obtained within R1 from 20 to 2, thus more ${\mathrm{HO}}^{•}$ radicals were generated. Figure 6b shows that within the first 5 min a higher concentration of H_2_O_2_ was consumed with the application of R1 = 2, which is consistent with the higher concentration of iron present, thus more ${\mathrm{HO}}^{•}$ radicals were generated. However, from this point onwards, the H_2_O_2_ consumption decreased to values lower than those observed with the application of R1 = 4, which could be a strong indicator that Fe^2+^ began consuming the ${\mathrm{HO}}^{•}$ radicals. To confirm this theory, the Fe^2+^ present in the solution was determined. The results showed higher Fe^2+^ concentrations available within the application of R1 from 20 to 2 (Figure 6b). A similar behavior was observed in the work of Faggiano et al. [45], in which increasing the concentration of iron to a certain extent lead to a reduction of the H_2_O_2_ consumption.

The results obtained showed a good fitting to the PFO kinetic model, and by plotting the *k* values vs. [Fe^2+^], two different dominions are shown (Figure 7a). In a first dominion, up to 1.1 × 10^−4^ mol L^−1^, the PFO kinetic constant is directly proportional to the concentration of Fe^2+^, with a slope of 0.071 mol L^−1^ min^−1^. A second dominion was observed from 1.1 × 10^−4^ to 11 × 10^−4^ mol L^−1^, in which the PFO kinetic constant increased linearly with the concentration Fe^2+^, with a slope of 0.749 mol L^−1^ min^−1^. In a closer look, from 5.5 × 10^−4^ to 11 × 10^−4^ mol L^−1^, the kinetic rate decreased. This could be attributed to scavenging reactions between the excess of iron present in solution and the ${\mathrm{HO}}^{•}$ radicals generated (Equation (11)) [46].
(11)$$\mathrm{Fe}2^{+}+{\mathrm{HO}}^{•}\;\to \;\mathrm{Fe}3^{+}+{\mathrm{HO}}^{-}$$

These results were in agreement with the work of Xavier et al. [47], who observed that the application of an excess of Fe^2+^ increased the scavenging reactions and decreased the efficiency of the photo-Fenton process in the degradation of Magenta MB. Parallel to the kinetic rate, the energy consumption was determined to evaluate the feasibility of the reactor. The results (Figure 7b) showed two different dominions, one for [Fe^2+^] ≤ 1.1 × 10^−4^ mol L^−1^ and one for [Fe^2+^] ≥ 1.1 × 10^−4^ mol L^−1^. These results clearly stipulate that for concentrations below 1.1 × 10^−4^ mol L^−1^, the reactions were too prolonged, implying larger electric consumptions, thus it could result in higher costs for treatment plants. For application of Fe^2+^ above 1.1 × 10^−4^ mol L^−1^, the gains were very low and do not justify the application of those amounts of iron. The selection of the appropriate radiation type with the correct geometry influences the kinetic rate of degradation and the costs of treatment. This fact was observed in the work of Tapia-Tlatelpa et al. [48], who observed from testing UV lamps with different geometries, that LED lamps with a radial position (similar to this work) achieved the lowest energy consumption.

In Table 2, it is shown research that compares the $E_{\mathrm{EO}}$ values obtained by other treatment processes applied in the degradation of phenolic compounds, dyes, and compounds of emerging concern. When compared to the work of Yáñez et al. [12], the methodology created in this work was shown to be more efficient, reducing the energy required to degrade the CA phenolic. In a different work, the application of a low-pressure Hg lamp (8 W power, 250 nm) presented an ideal process to convert the H_2_O_2_ into ${\mathrm{HO}}^{•}$ radicals [49]; however, without a catalyst, this was observed to be a slow process with high energy consumption.

In the work of Kim et al. [50], three different reactors were applied: (1) UV-C lamp (6 W power, 254 nm) with H_2_O_2_, (2) ozone batch reactor (1.12 mg O_3_/min), and (3) E-beam (1 MeV and 40 kW). The results showed that the ozonation process achieved worse $E_{\mathrm{EO}}$ results in comparison to the UV-A reactor used in this work. Although the UV-C and E-beam reactors showed promising results, the concentration of the SMX degraded was much lower than the concentration of CA used in this work. The UV-A reactors with an emission wavelength of 365 nm were applied in photocatalytic degradation of dyes and phenolics [33,48], with results showing efficient removals; however, they are not revealed as being competitive in comparison to the results obtained in this work. In Table 2, several examples of the application of photo-Fenton process to treat real wastewaters are also shown, such as in winery wastewater [51] and poultry slaughterhouse wastewater [52]. It is shown that the demand for electric energy to treat real wastewaters is much higher in comparison to single compounds, as is the amount of reagents required to degrade the organic carbon.

### 3.5. Experiments in Continuous Mode

Previous sections have shown that batch reactors fitted with UV-A LED lights achieved a near complete removal of CA from the aqueous solution. It was also observed in the H_2_O_2_ and Fe^2+^ concentration variations sections that the degradation of CA occurred in two phases, a faster one within the first 5 min, followed by a slower rate, reaching a plateau. This occurs because, in the first period, the Fe^2+^ reacted very fast with the H_2_O_2_ (k = 78 mol^−1^ dm^3^ s^−1^), generating a high content of ${\mathrm{HO}}^{•}$ radicals. Following this period, the ferric ions produced earlier reacted with the H_2_O_2_ producing ${\mathrm{HO}}_{2}^{•}$, with a much lower kinetic rate (k = 0.02 mol^−1^ dm^3^ s^−1^) [55]. Therefore, these results indicate that the residence time (τ, min) within the continuous mode is an important parameter to be considered. In an ideal continuous reactor, the space-time is equal to the mean residence time, and is determined in accordance with Equation (12) [55].
(12)$$\tau =\frac{V}{F}$$
where V is the volume of the reactor (mL) and *F* is the flow rate (mL min^−1^). In Figure 8a are shown the removal results of CA and DOC as a function of dimensionless reaction time values (θ = t/τ), in which each θ is comparable to one τ [56]. The results showed that a CA steady state was achieved after 0.04 space-time values, while the mineralization of CA showed a steady state from 0 to 0.020 space-time values. By the analysis of the results obtained, the flow rate had a significant influence on the removal rate of CA and mineralization, in which, with *F* = 4 mL min^−1^, higher CA and DOC removal was achieved (99.2 and 35.9%, respectively). The Fe^2+^ concentration analysis revealed that the availability of iron remaining in the solution was similar to that observed in batch experiments; therefore, the concentration of H_2_O_2_ was revealed to be the decisive factor. In observation of Figure 8, it can be seen that the application of *F* = 1 and 2 mL min^−1^ led to considerable lower additions of H_2_O_2_ regarding the application of *F* = 4 mL min^−1^, thus less ${\mathrm{HO}}^{•}$ radicals were generated. These results were in agreement with the work of Rosales et al. [30], who observed that increasing the residence time enhanced the generation of ${\mathrm{HO}}^{•}$ radicals, increasing the discoloration of textile wastewaters.

To understand the results obtained by the continuous mode, the kinetic constants were determined by application of a continuous mode simplified model, assuming a first order reaction Equation (13) [33].
(13)$$k\tau =\frac{1-\;\frac{C_{\mathrm{ss}}}{C_{0}}}{\frac{C_{\mathrm{SS}}}{C_{0}}}$$
where C_0_ and C_SS_ are the initial and steady state concentrations of CA (mol L^−1^) and DOC (mg C L^−1^). In Table 3 are shown the kinetic rate constants obtained under different flow rates. The results are plotted in Figure 9, showing that as the flow rate increases, the kinetic rate of CA and DOC also increased, meaning that it achieved a good penetration of the UV-A radiation into the water. Considering this behavior, the continuous mode in association with the UV-A LEDs with horizontal geometry could be considered as a good system for phenolic degradation.

## 4. Conclusions

This work attempted to study the effect of several parameters in the kinetic rate and energy consumption of the hydroxyl radical-based oxidation processes. The experiments were carried out employing a UV-A LED system emitting at 365 nm, H_2_O_2_ as the oxidant, and Fe^2+^ as the catalyst. Based in the results, the UV-A-Fenton process at pH 3.0 achieved the highest CA degradation rate. It is concluded that increasing the H_2_O_2_ concentration increases the kinetic rate of CA degradation; however, above 2.2 × 10^−3^ mol L^−1^, the reactor is not economically viable, because no significant $E_{\mathrm{EO}}$ reduction is observed. It is concluded that the Fe^2+^ achieves the best performance, with 1.1 × 10^−4^ mol L^−1^, with an $E_{\mathrm{EO}}$ = 7.23 kWh m^−3^ order. Above this concentration, the kinetic rate increases; however, no significant gains in energy consumption were observed. When batch conditions are applied to continuous mode, it is concluded that the degradation of CA is dependent on the flow rate. It is also concluded that higher flow rates lead to higher CA and DOC removals (k_CA_ = 0.468 min^−1^; k_DOC_ = 0.00896 min^−1^).

Based in the results obtained from this work, a UV-A LED reactor adapted to continuous mode could provide an excellent process to reduce the soaring energy costs associated with these systems and make them viable for wastewater treatment plants.

## Figures and Tables

**Figure 1 ijerph-20-01276-f001:**
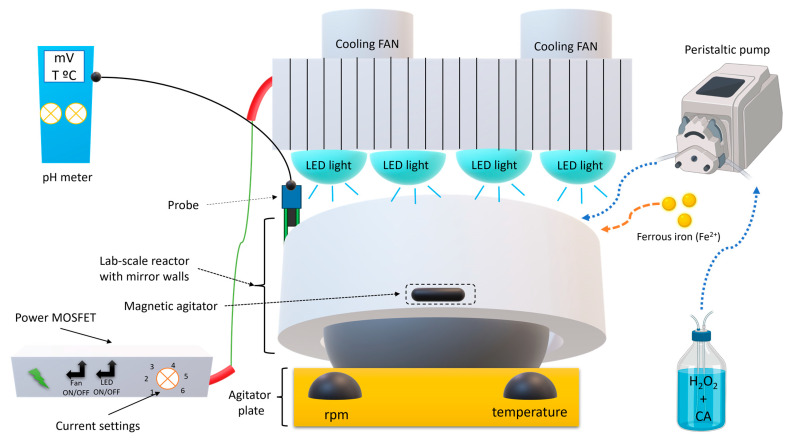
Schematic representation of UV-A LEDs lab-scale reactor with peristaltic pump.

**Figure 2 ijerph-20-01276-f002:**
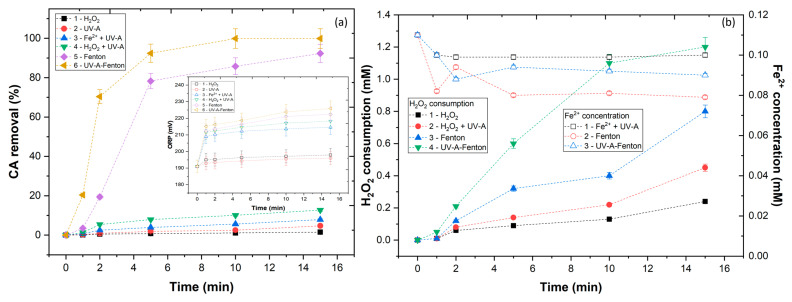
Assessment of different AOPs in (**a**) CA removal and ORP variation and (**b**) H_2_O_2_ consumption and Fe^2+^ concentration. Operational conditions: [CA] = 5.5 × 10^−4^ mol L^−1^, [H_2_O_2_] = 2.2 × 10^−3^ mol L^−1^, [Fe^2+^] = 1.1 × 10^−4^ mol L^−1^, pH = 3.0, agitation 150 rpm, temperature = 298 K, radiation UV-A, $I_{\mathrm{UV}}$ = 32.7 W m^−2^, t = 15 min.

**Figure 3 ijerph-20-01276-f003:**
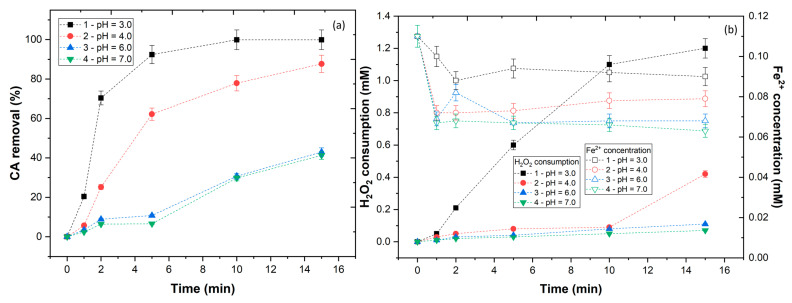
Effect of pH in (**a**) CA removal and (**b**) H_2_O_2_ consumption and Fe^2+^ concentration. Operational conditions: [CA] = 5.5 × 10^−4^ mol L^−1^, [H_2_O_2_] = 22 × 10−4 mol L^−1^, [Fe^2+^] = 1.1 × 10−4 mol L^−1^, agitation 150 rpm, temperature = 298 K, radiation UV-A, $I_{\mathrm{UV}}$ = 32.7 W m^−2^, t = 15 min.

**Figure 4 ijerph-20-01276-f004:**
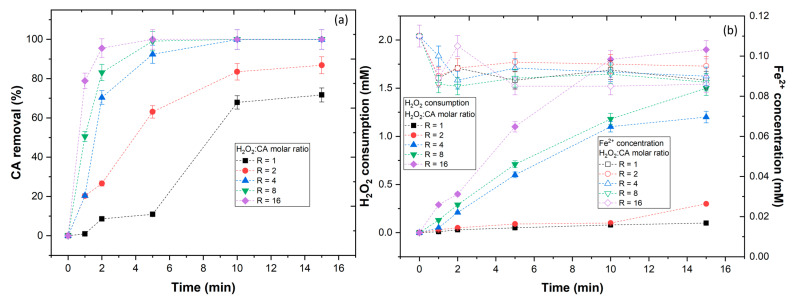
Evaluation of H_2_O_2_:CA molar ratio in (**a**) CA removal and (**b**) H_2_O_2_ consumption and Fe^2+^ concentration. Operational conditions: [CA] = 5.5 × 10^−4^ mol/L, [Fe^2+^] = 1.1 × 10^−4^ mol L^−1^, pH = 3.0, agitation 150 rpm, temperature = 298 K, radiation UV-A, $I_{\mathrm{UV}}$ = 32.7 W m^−2^, t = 15 min.

**Figure 5 ijerph-20-01276-f005:**
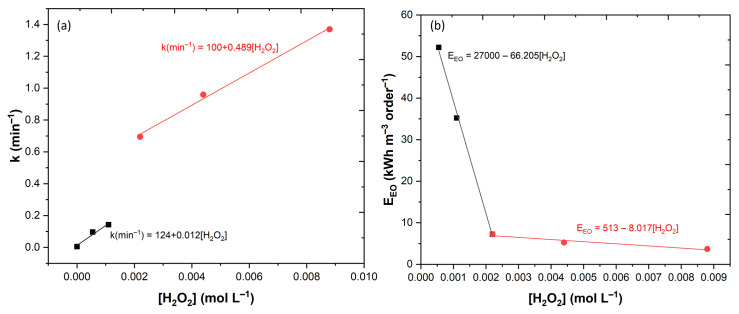
Outcome of several concentrations of H_2_O_2_ in CA degradation by the UV-A-Fenton process; (**a**) dependence on the kinetic constant rates; (**b**) electric energy per order ($E_{\mathrm{EO}}$).

**Figure 6 ijerph-20-01276-f006:**
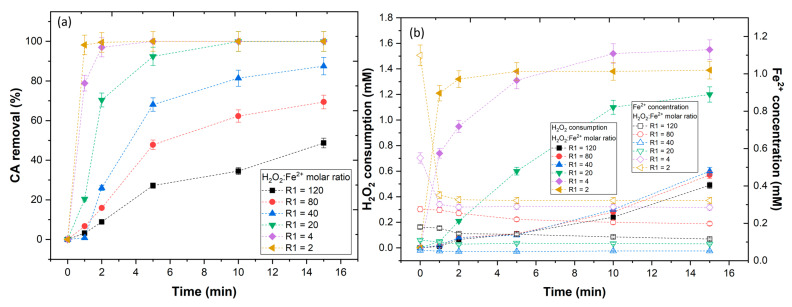
Assessment of the H_2_O_2_:Fe^2+^ molar ratio in (**a**) CA degradation and (**b**) H_2_O_2_ consumption and Fe^2+^ concentration. Operational conditions: [CA] = 5.5 × 10^−4^ mol L^−1^, [H_2_O_2_] = 2.2 × 10^−3^ mol L^−1^, pH = 3.0, agitation 150 rpm, temperature = 298 K, radiation UV-A, $I_{\mathrm{UV}}$ = 32.7 W m^−2^, t = 15 min.

**Figure 7 ijerph-20-01276-f007:**
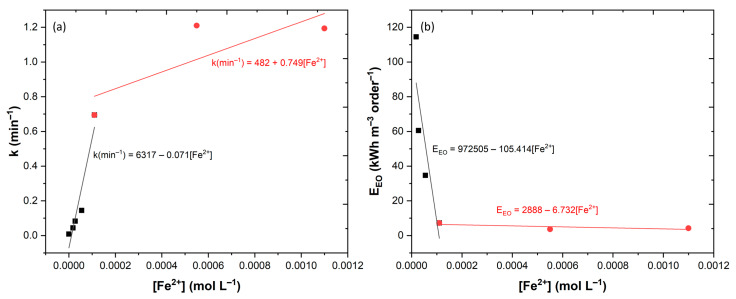
Outcome of several concentrations of Fe^2+^ in CA degradation by UV-A-Fenton process; (**a**) dependence on the kinetic constant rates; (**b**) electric energy per order ($E_{\mathrm{EO}}$).

**Figure 8 ijerph-20-01276-f008:**
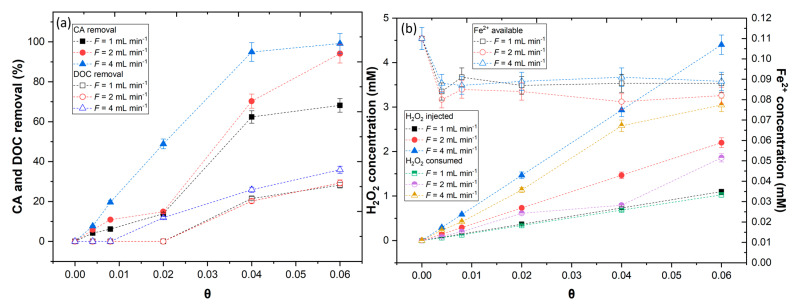
Assessment of different flow rates (1–4 mL min^−1^) in (**a**) CA and DOC removal; (**b**) H_2_O_2_ and Fe^2+^ concentration available as function of θ. Operational conditions: [CA] = 5.5 × 10^−4^ mol L^−1^, [Fe^2+^] = 1.1 × 10^−4^ mol L^−1^, pH = 3.0, agitation 150 rpm, temperature = 298 K, radiation UV-A, $I_{\mathrm{UV}}$ = 32.7 W m^−2^, HRT = 15 min.

**Figure 9 ijerph-20-01276-f009:**
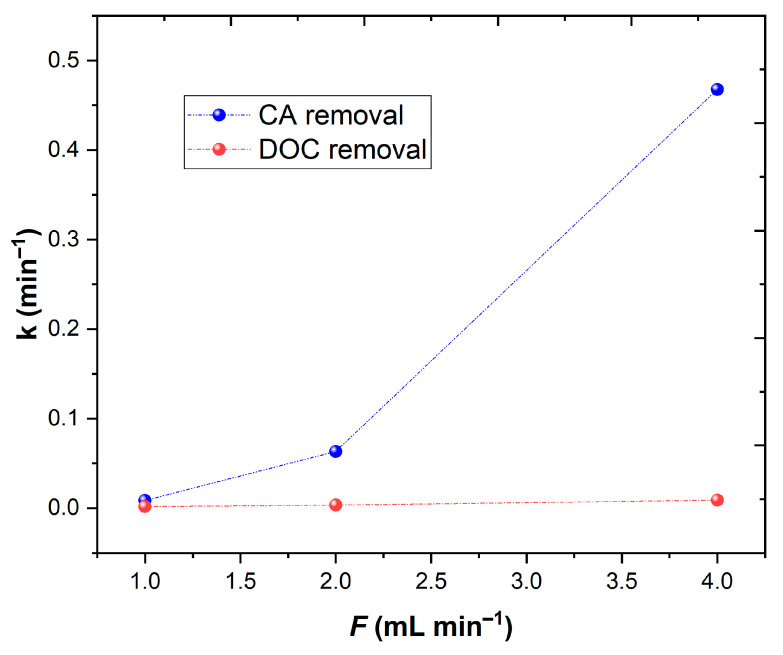
Dependence of the kinetic constant (k) to the flow rate (*F*) of the continuous mode reactor.

**Table 1 ijerph-20-01276-t001:** Molecular formula, molecular structure, maximum absorption wavelength and molecular weight of caffeic acid (CA) [31].

Name	Molecular Formula	Molecular Structure	λ_max_ (nm)	Molecular Weight (g/mol)
Caffeic acid (CA)	(HO)_2_C_6_H_3_CH=CHCO_2_H	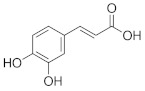	324	180.16

**Table 2 ijerph-20-01276-t002:** Disclosed results of electric energy per order ($E_{\mathrm{EO}}$ ) obtained after degradation of single compounds by different treatment processes.

Contaminants	AOP Processes	Conditions	*E*_EO_ (kWh.m^−3^ order^−1^	References
Caffeic acid	UV-A LED/H_2_O_2_/Fe^2+^	[CA] = 5.5 × 10^−4^ mol L^−1^, [H_2_O_2_] = 2.2 × 10^−3^ mol L^−1^, [Fe^2+^] = 1.1 × 10^−4^ mol L^−1^	7.23	Present work
Caffeic acid (CA)	UV-A-Fenton	[CA] = 10 mg L^−1^, [H_2_O_2_] = 82.4 µmol L^−1^, [Fe^2+^] = 558.6 µmol L^−1^	30	Yáñez et al. [12]
Winery wastewater	UV-A-Fenton	TOC = 1601 mg C L^−1^, [Fe^2+^] = 2.5 mM, [H_2_O_2_] = 225 mM, pH = 3.0, agitation = 350 rpm, t = 150 min	641	Jorge et al. [51]
Winery wastewater	UV-C-Fenton	TOC = 1601 mg C L^−1^, [Fe^2+^] = 2.5 mM, [H_2_O_2_] = 225 mM, pH = 3.0, agitation = 350 rpm, t = 150 min	170	
Poultry slaughterhouse wastewater	UV-C-Fenton	TOC = 68.66 mg C L^−1^, [Fe^2+^] = 20 mg L^−1^, [H_2_O_2_] = 98 mM, pH = 3.3, agitation = 350 rpm, t = 150 min	248	Kanafin et al. [52]
Oxytetracycline (OTC)	UV-C/H_2_O_2_	[OTC] = 250 mg L^−1^, [H_2_O_2_] = 375 mg L^−1^	47.18	Rahmah et al. [49]
Sufamethoxazole (SMX)	UV-C	[SMX] = 30 mg L^−1^, [H_2_O_2_] = 10 mM	1.50	Kim et al. [50]
Sufamethoxazole (SMX)	Ozone	[SMX] = 30 mg L^−1^	27.53	
Sufamethoxazole (SMX)	Electron beam	[SMX] = 30 mg L^−1^	0.46	
Acid Red 88 (AR88)	UV-A-Fenton	pH 3.0, [AR88] = 50 mg L^−1^, [H_2_O_2_] = 4 mM, [Fe^2+^] = 0.15 mM, [NTA] = 0.10 mM,	26	Teixeira et al. [53]
Orange PX-2R	UV-A LED/TiO_2_	[OPX-2R] = 0.1 g/L, [TiO_2_] = 1.0 g/L	119.04	Tapia-Tlatelpa et al. [48]
*p*-hydroxybenzoic acid (pHBA)	UV-A LED/TiO_2_	[pHBA] = 50 mg L^−1^, [TiO_2_] = 1000 mg L^−1^	115	Ferreira et al. [33]
Z-thiacloprid	UV/TiO_2_	[Z-thiacloprid] = 1.0 × 10^−4^ mol L^−1^, [TiO_2_] = 1 g L^−1^	80.0	Rózsa et al. [54]

**Table 3 ijerph-20-01276-t003:** Kinetic constants from CA and DOC removal observed for the assays with different flow rates (mL min^−1^). Means in the same column with different letters represent significant differences (*p* < 0.05) within each parameter (k_CA_ and k_DOC_) by comparing flow rates.

F	τ	C_SS_/C_0_	DOC_SS_/DOC_0_	k_CA_	k_DOC_
mL min^−1^	min			min^−1^	min^−1^
1	250	0.318	0.718	0.009 ± 4.5 × 10^−4 a^	0.00157 ± 7.8 × 10^−5 a^
2	125	0.059	0.707	0.063 ± 3.2 × 10^−3 b^	0.00332 ± 1.7 × 10^−4 b^
4	62.5	0.008	0.641	0.468 ± 2.3 × 10^−2 c^	0.00896 ± 4.5 × 10^−4 c^

## Data Availability

Not applicable.

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
