# Peer review of "Degradation of Agro-Industrial Wastewater Model Compound by UV-A-Fenton Process: Batch vs. Continuous Mode"

_ijerph, 2023, doi:10.3390/ijerph20021276_

Round 1
Reviewer 1 Report
The author reported, “Degradation of agro-industrial wastewater model compound by UV-A-Fenton process: batch vs continuous mode”. The authors discussed the possible mechanisms of this work. The following comments must be addressed before acceptance.
1. In Fig. 5b, can the H2O2 consumption of R1=4 exceeds R1=2 in 6min be explained?
2. In Table 2, apart from this work, no other studies were done on CA degradation. There is also only one study related to photo-Fenton. Can more relevant studies be listed?
3. The units in Table 2 could ideally be standardised.
Author Response
Dear Reviewer, the authors thank you for your important comments, which will enhance the quality of this work. The changes were highlighted in yellow.

Reviewer 2 Report
The degradation of a model agro-industrial wastewater phenolic compound (caffeic acid, CA) by a UV-A-Fenton system was investigated in this work. Experiments were carried out in order to compare batch and continuous mode. Initially, batch experiments shown that UV-A-Fenton at pH 3.0 (pH of CA solution) achieved higher generation of HO• leading to high CA degradation (>99.5%). The influence of different operational conditions, such as H2O2 and Fe2+ concentrations were evaluated. The results fit a pseudo first-order (PFO) kinetic model and a high kinetic rate of CA removal was observed, the present work is practical and useful.
The authors may consider following minor revisions.
1. Please mention the actual CA concentrations in wastewater along its impacts on the ecological phenomenon. What is half life of CA in natural conditions?
2. Research design needs improvements. How much CA was used in various runs, what was HRT, what was optimum reaction time, why H2O2 was used?
3. Please add a statistical analysis of the results.
Author Response

(The authors gave the same response as above.)

Author Response

(The authors gave the same response as above.)
